# The Role of GREMLIN1, a Bone Morphogenetic Protein Antagonist, in Cancer Stem Cell Regulation

**DOI:** 10.3390/cells14080578

**Published:** 2025-04-11

**Authors:** Yuhan Gao, Swapnali De, Derek P. Brazil

**Affiliations:** Wellcome-Wolfson Institute for Experimental Medicine, School of Medicine, Dentistry, and Biomedical Sciences, Queen’s University Belfast, 97 Lisburn Road, Belfast BT9 7BL, Northern Ireland, UK; ygao19@qub.ac.uk (Y.G.); sde01@qub.ac.uk (S.D.)

**Keywords:** GREMLIN1, bone morphogenetic protein, antagonist, cancer stem cell, pluripotency

## Abstract

Cancer remains a leading cause of death globally, characterized by uncontrolled cell proliferation, tumor growth and metastasis. Bone morphogenetic proteins (BMPs) and their growth differentiation factor (GDF) relatives are crucial regulators of developmental processes such as limb, kidney and lung formation, cell fate determination, cell proliferation, and apoptosis. Cancer stem cells (CSCs) are a subpopulation of self-renewing cells within tumors that possess stemness properties and a tumor cell-forming capability. The presence of CSCs in a tumor is linked to growth, metastasis, treatment resistance and cancer recurrence. The tumor microenvironment in which CSCs exist also plays a critical role in the onset, progression and treatment resistance in many cancers. Growth factors such as BMPs and GDFs counterbalance transforming growth factor-beta (TGF-β) in the maintenance of CSC pluripotency and cancer cell differentiation. BMP signaling typically functions in a tumor suppressor role in various cancers by inducing CSC differentiation and suppressing stemness characteristics. This differentiation process is vital, as it curtails the self-renewal capacity that characterizes CSCs, thereby limiting their ability to sustain tumor growth. The interplay between BMPs and their secreted antagonists, such as GREM1, Noggin and Chordin, adds another layer of complexity to CSC regulation. Human cancers such as gastric, colorectal, glioblastoma, and breast cancer are characterized by GREMLIN1 (GREM1) overexpression, leading to inhibition of BMP signaling, facilitating the maintenance of pluripotency in CSCs, thus promoting tumorigenesis. GREM1 overexpression may also contribute to CSC immune evasion, further exacerbating patient prognoses. In addition to BMP inhibition, GREM1 has been implicated as a target of fibroblast growth factor (FGF) → Sonic hedgehog (Shh) signaling, as well as the Wnt/Frizzled pathway, both of which may contribute to the maintenance of CSC stemness. The complex role of BMPs and their antagonists in regulating CSC behavior underscores the importance of a balanced BMP signaling pathway. This article will summarize current knowledge of BMP and GREM1 regulation of CSC function, as well as conflicting data on the exact role of GREM1 in modulating CSC biology, tumor formation and cancer. Targeting this pathway by inhibiting GREM1 using neutralizing antibodies or small molecules may hold early-stage promise for novel therapeutic strategies aimed at reducing CSC burden in cancers and improving patient outcomes.

## 1. Introduction

Cancer remains a significant global health challenge, with approximately 20 million new cases and 9.7 million deaths in 2022, according to the International Agency for Research on Cancer (IARC) [1]. The most common forms of cancer in the UK are breast, prostate, lung, and intestinal cancer, accounting for more than 50% of the cancer diagnoses in 2021 [2]. A 5% increase in cancer diagnoses in the UK was reported in 2022, which may reflect earlier and improved detection, as well as an aging population [3]. Colorectal cancer (CRC) ranks as the third most commonly diagnosed cancer worldwide (1.9 million cases, 9.6%) and the second leading cause of cancer-related deaths (0.9 million cases, 9.3%), underscoring its critical impact on public health and clinical resources [1]. Tumor initiation and progression in cancers of the breast, pancreas, prostate, lung, and intestine and other tissues are driven by several factors such as mutations in oncogenes like KRAS (KRAS^G12D^), loss of tumor suppressors such as p53, adenomatous polyposis coli (APC), and the PTEN lipid and protein phosphatase [4,5,6,7]. Tumor heterogeneity can alter the response to chemotherapy and recurrence, which can also be influenced by the presence of cancer stem cells (CSCs) [8]. The ability of CSCs to maintain their quiescent/pluripotent state involves complex signaling networks, among which the bone morphogenetic protein (BMP) pathway plays a pivotal role. BMP2 has been shown to promote CSC differentiation in glioma and other cancers [9]. Dysregulation of BMP signaling is often observed in cancers, and can lead to dysregulated CSC differentiation, promoting tumorigenesis [10]. GREM1, a BMP antagonist, has emerged as a key regulator of cancer development and progression. Overexpression of GREM1 has been linked to tumor progression in a range of cancers, where it is associated with enhanced stemness, increased metastasis, resistance to therapy, and poorer patient prognosis [11,12]. While GREM1-mediated inhibition of BMP signaling is likely to play a major role in the regulation of CSC quiescence and differentiation, other GREM1 signaling pathways may also play a role in CSC regulation. A summary of our current understanding of GREM1 biology in CSC regulation will be provided in this article.

## 2. Characteristics of Cancer Stem Cells

CSCs exhibit several features that distinguish them from the bulk of tumor cells. One of the defining characteristics of CSCs is their self-renewal ability. Like normal stem cells, CSCs can divide and produce identical daughter cells or differentiate into specialized cell types, contributing to tumor heterogeneity [13]. This diversity contributes to the complexity of tumors and is a significant factor in treatment resistance [14]. In addition to self-renewal, CSCs are often characterized by their ability to initiate and propagate tumors. In xenograft models, where human tumors are implanted in immunocompromised mice, only a small subset of cells (CSCs) can give rise to new tumors, supporting their pivotal role in cancer initiation and progression [15]. Additionally, CSCs tend to express specific markers, such as CD133, CD44, CD24, Thy1 and ALDH, which can aid in their identification [16,17]. However, many of these markers are also expressed by non-cancer cells, so caution is needed when making definitive conclusions in identifying CSCs. Since CSCs are predicted to be quiescent and have a lower rate of cell cycling, bromodeoxyuridine (BrdU) label retention has also been used in their identification [18]. Genetic and epigenetic signatures of CSCs have also been identified. A recent study identified the expression of Enhancer of Zeste homolog 2 (EZH2) in sarcoma CSCs that contributed to CSC resistance to doxorubicin chemotherapy [19]. Targeting EZH2 with the selective inhibitor tazemetostat reduced sarcoma CSC survival, demonstrating that insights into the genetic signatures of CSCs may provide novel therapeutic opportunities to target these cells [19]. Histone acetylation at H3K4 and H3K27 have also been shown to contribute to the maintenance of CSC quiescence in colorectal, glioma, and breast cancer [20,21,22,23]. The nuclear factor I X (NFIX) was discovered to regulate chromatin access and maintain CSC identity [24], and high levels of NFIX expression in gastric cancer were associated with increased tumor stemness and poorer patient prognosis [25]. Thus, a complex interplay of genetic and epigenetic mechanisms exists to regulate CSC biology in cancer.

One of the primary reasons why CSCs are of significance in cancer research is their role in chemotherapy resistance. Traditional therapies are designed to target rapidly dividing cancer cells, which form the bulk of solid tumors. However, CSCs often exist in a quiescent state, making them less susceptible to conventional chemotherapy regimens. As a result, after the bulk of the tumor is eliminated by chemotherapy, the surviving CSCs can initiate tumor recurrence and metastasis [26]. Additionally, CSCs may exhibit unique drug resistance mechanisms, such as increased expression of drug efflux pumps or anti-apoptotic factors [27,28]. Understanding these mechanisms is critical for developing more effective therapies to target CSCs to improve clinical outcomes for cancer patients.

## 3. Regulation of BMP Signaling by GREM1

The role of BMP signaling in cancer stemness and differentiation has recently been reviewed in detail [10]. Roles for BMP signaling in human cancers include the promotion of differentiation, migration, and invasion in glioblastoma, prostate, and colorectal cancer, reduced stemness in osteosarcoma and the promotion of migration and invasiveness in lung cancer [10]. GREM1 is one of a DAN family of secreted cysteine knot extracellular antagonists that target BMP pathways. Other BMP antagonists such as Noggin, Chordin, and GREM2 also regulate BMP signaling, with differing tissue-specific expression and affinities for individual BMPs evident among these BMP antagonists [29]. GREM1 (and other antagonists) bind to BMPs in the extracellular matrix, preventing BMP interaction with cognate BMPR1 (ALK1/2/3/6)/BMPR2 receptor complexes, inhibiting the activation of BMP receptor signaling [30,31]. When BMP dimers engage these receptors, it leads to phosphorylation of the receptor, thereby activating it. This, in turn, phosphorylates R-Smads (SMAD1/5/8) that then form a complex with the co-SMAD4 [32]. This SMAD complex then translocates to the nucleus where it binds to transcriptional complexes at BMP-response elements in a specific subset of genes such as inhibitor of differentiation-1 (ID1) and the inhibitory SMAD7 [33]. These BMP-regulated genes contribute to the regulation of immune responses, apoptosis, cell cycle regulation, cell differentiation and organ development [34].

Tight temporospatial control of GREM1 and BMP signaling is essential for physiological development of lower limbs, kidneys and other tissues and organs during embryogenesis. GREM1 also plays an important role in epithelial homeostasis in the lung and intestine. GREM1 expression has also been identified as a marker of osteochondroreticular (OCR) stem cells in the bone marrow [35]. These OCR cells can self-renew and differentiate into osteoblasts, chondrocytes and reticular marrow stromal cells, contributing to bone development, remodeling and fracture repair [35]. Loss of these GREM1-lineage chondrogenic progenitor cells has been linked to osteoarthritis in mice, involving perturbations in FGF18 → FGFR3 signaling [36]. These data and others position GREM1 as a key mediator of stem cell function in normal, physiological development. The interplay between GREM1 and BMP targets is summarized in Figure 1.

### 3.1. GREM1 Signaling in Cancer

In healthy tissues, GREM1 mRNA levels are generally low, with the highest levels detected in the GI tract, gall bladder, smooth muscle and adipose tissue [37]. In adulthood, upregulation of GREM1 expression has been linked to a range of human cancers. Elevated levels of GREM1 mRNA were detected in human basal cell carcinoma (BCC) stromal cells [38]. GREM1 overexpression was associated with metastasis in breast cancer and was associated with poorer patient prognosis [39], GREM1 expression was detected in cancer-associated fibroblasts (CAFs) promoting breast cancer extravasation in a Zebrafish model [40]. Elevated GREM1 signaling in CAF subtypes in cancer contributes to epithelial–mesenchymal transition and the formation of mesenchymal cancer cells, increased solid tumor formation, invasion and metastasis [41]. Loss of H3K27 methylation of the GREM1 gene has been identified as one potential mechanism of GREM1 mRNA upregulation in CAFs in gastric cancer [42]. Overexpression of GREM1 has been described in epithelial cells in a genetic condition called hereditary mixed polyposis syndrome (HMPS), as well as the more common sporadic colorectal serrated adenoma. The effect of GREM1 overexpression is thought to primarily be excessive inhibition of BMP signaling, but other non-BMP signaling mechanisms may also result from excessive GREM1 expression in disease.

The exact mechanisms driving GREM1 overexpression in cancer cells are not fully understood. Many groups have identified single nucleotide polymorphisms (SNPs) in the GREM1 gene locus that are associated with upregulated GREM1 mRNA production. For example, Tomlinson and colleagues were the first to identify an association between tagSNPs at the *GREM1* genomic locus (15q13.3) and CRC [43]. This group also identified a common SNP (rs16969681) close to the *GREM1* gene that increases GREM1 mRNA expression due to increased affinity of transcription factors TCF7L2 and CDX2 for a *GREM1* enhancer region [44]. Fortini and colleagues identified a CRC risk-associated SNP (rs4779584) in an enhancer region of 15q13.3 that regulates GREM1 [45]. Changes in microRNAs have also been implicated as a potential mechanism of GREM1 upregulation in cancer. For example, a low frequency variant rs12915554 was identified in the 3’ UTR of *GREM1* that was significantly associated with CRC risk [46]. These authors demonstrated that the presence of this polymorphism disrupted the binding of miR-185-3p, thus preventing miRNA-mediated repression of GREM1 mRNA expression [46]. A more recent report identified that miR-1236-3p could bind to GREM1 mRNA and inhibit its expression [47]. Intriguingly, levels of a circular RNA (or competing endogenous RNA (ceRNAs)) called circ_0000212 that directly binds to and inhibits miR-1236-3p was shown to enhance GREM1 expression, increasing cervical cancer risk [47]. It is likely that additional regulatory mechanisms such as transcription factor overexpression and binding to the *GREM1* promoter, additional epigenetic modification of the *GREM1* gene locus, GREM1 mRNA stability/localization and GREM1 protein posttranslational modification may also contribute to sustained GREM1 expression in cancer.

In contrast to the overwhelming volume of publications implicating GREM1 as a “bad actor” in cancer, a small number of papers argue the opposite. For example, a study in 2013 identified that GREM1 expression correlated with increased angiogenesis and progression-free survival in patients with pancreatic neuroendocrine tumors, suggestive of a tumor suppressor role for GREM1 [48]. A more recent study focused on GREM1 in pancreatic ductal carcinoma (PDAC) and demonstrated that GREM1 overexpression drove the epithelialization of mesenchymal PDAC tumors [49]. This effect was linked to suppression of the Snail/Slug transcription factors that drive epithelial–mesenchymal transition (EMT), suggesting that high GREM1 expression restricted epithelial–mesenchymal plasticity in pancreatic cancer [49]. Despite the overwhelming evidence in the literature that GREM1 is a tumor promoter in cancer, the conflicting data on its role as a tumor suppressor muddies the waters somewhat, and should be carefully considered [41].

### 3.2. GREM1 Regulation of CSC Stemness Maintenance

One of the fundamental properties of CSCs is the maintenance of specific signaling pathways that promote their self-renewal and reduce their differentiation. GREM1 has been shown to influence a number of these pathways, specifically the BMP, TGF-β, and Wnt signaling pathways. BMP2 and BMP4 have been identified as key inhibitors of CSC self-renewal and drivers of CSC differentiation, for example in glioblastoma (GBM) stem cells [9]. Transcription factors such as SNAIL and DLX2 have been implicated in this tumor suppressor-like effect downstream of BMPs [50,51]. Increased levels of BMP2 (which drives CSC differentiation toward post-mitotic cancer cells) have been detected in GBM tumors; however, these tumors still retain a subpopulation of pluripotent CSCs, despite high BMP2 expression [9]. The explanation for this phenomenon is that these GBM tumors also secrete high levels of GREM1, which binds to and inhibits BMP2 signaling, thereby limiting the ability of BMP2 to induce CSC differentiation [9]. These effects of GREM1 were linked to the inhibition of the universal cyclin-dependent kinase inhibitor-1 p21 (WAF1/CIP1), an essential signaling pathway for CSCs. This inhibition of BMP signaling via excessive GREM1 levels has been associated with the promotion of stem cell characteristics, leading to enhanced tumorigenicity. GREM1 mRNA levels are also elevated in cervical CSCs and associated with poorer patient prognosis [52]. GREM1 is secreted from mesenchymal stromal cells (MSCs) and enhanced the malignancy of xenograft esophageal tumors in vivo, an effect linked to GREM1-induced EMT in esophageal squamous cell carcinoma [53].

GREM1 expression in cancer is also associated with a fibrogenic, pro-epithelial to mesenchymal transition (EMT)-like environment. GREM1 has been reported to enhance TGF-β-induced extracellular matrix protein production linked to an EMT-like phenotype in pancreatic ductal carcinoma (PDAC) [54]. Knockdown of GREM1 expression blunted TGF-β-induced fibronectin, collagen I and alpha-smooth muscle actin (α-sma) expression [54]. GREM1 delivered by mesenchymal stem cells (MSCs) was overexpressed in esophageal squamous cell carcinoma (ESCC), leading to enhanced EMT via TGF-β/BMP signaling [53]. siRNA-mediated knockdown of GREM1 in U87-MG glioblastoma cells induced cell cycle arrest and apoptosis, and abolished TGF-β1-mediated activation of SMAD2 signaling, inhibiting EMT in these cells [55]. These sample reports suggest intricate paracrine signaling loops where GREM1 and other proteins are secreted from MSCs and other cells, and act on differing cell types in the tumor microenvironment. These data also highlight somewhat conflicting data on the ability of GREM1 to modulate TGF-β signaling, contributing to the maintenance of the CSC pluripotent phenotype and increasing the aggressiveness of tumors.

During development, regulation of GREM1 expression by fibroblast growth factor (FGF) and sonic hedgehog (Shh) was identified as a key network controlling limb bud outgrowth [56]. This report identified that an inhibitory loop exists whereby high levels of FGF act to repress GREM1 expression in stem cells of the developing limb bud [56]. Gli transcription factors (Gli1-3) downstream of Shh signaling were identified as key silencers of GREM1 expression [57]. In human cancer, GREM1 expression has been demonstrated in pancreatic stellate cells (PSCs), and is driven by Shh → Gli1 signaling, enabling pancreatic cancer progression [58]. GREM1 signaling via FGFR1 → MAPK has been identified as a key signaling pathway regulating lineage plasticity in castration-resistant prostate cancer [59]. This report identified that upregulation of GREM1 expression in prostate cancer was associated with increased stem cell-related gene expression. In addition, targeting GREM1 using a neutralizing antibody produced a tumor-inhibiting effect in a mouse model of castration-resistant prostate cancer [59]. Shh signaling and Gli transcription factors are aberrantly activated in glioma, eosphageal, and pancreatic cancer [60]. Future experiments defining the role of FGF, Shh and Gli transcription factors in GREM1 regulation in CSC biology will further expand our understanding of these complex signaling networks. These data are summarized in Figure 2.

GREM1 also regulates Wnt signaling, another crucial pathway in CSC regulation. Wnt signaling promotes the self-renewal and proliferation of CSCs in various cancers. The Wnt-regulated transcription factor 7-like 2 (TCF7L2) was shown to bind close to a polymorphism site in an enhancer for GREM1, increasing GREM1 expression [44]. Interactions between GREM1 and Wnt signaling have also been identified in the maintenance of Lgr4+ epithelial stem cells in human intestinal organoids [61]. By regulating Wnt signaling, GREM1 may further enhance the stem cell-like properties of CSCs (Figure 2). It has been reported that GREM1 can activate vascular endothelial growth factor receptor-2 (VEGFR2) signaling in endothelial cells to promote angiogenesis [62,63]. However, other researchers (including the author’s group) could not validate this GREM1 → VEGFR2 signaling pathway in endothelial cells [64]. VEGF signaling via VEGFR1-3 as well as neuropilin receptors NRP1 and NRP2 have been identified as key mediators of CSC self-renewal and resistance to chemotherapy [65]. It remains an intriguing possibility that GREM1 signaling via VEGFR2 may represent a previously undiscovered mechanism by which GREM1 regulates CSC pluripotency and chemotherapy resistance in human cancers.

It is important to recognize that other BMP antagonists such as Noggin and Chordin have also been implicated in cancer progression and metastasis. Noggin is an antagonist of BMP7 that regulates bone and heart formation [66]. Noggin regulates the ability of metastatic breast cancer cells to colonize bone tissue by promoting osteoclast differentiation and other mechanisms [67]. Others have highlighted that Noggin could inhibit BMP4-induced EMT and Notch gene expression in MCF-10A breast cancer cells [68]. Chordin-like 1 (CHRDL1) has been identified as an “enforcer of stemness” in glioma stem cells, via antagonism of BMP4-induced glioblastoma differentiation and reduced tumorigenicity [69]. Follistatin (FST) is a secreted antagonist of TGFβ1 that maintains epithelial cells in their progenitor state via inhibition of TGFβ1 signaling [70]. Data on FST function inCSCs are more limited, but some reports demonstrate that FST secretion from cancer-associated fibroblasts (CAFs) enhances CAF-mediated cancer cell proliferation and metastasis [71]. These data highlight the role of other BMP antagonists apart from GREM1 in CSC and cancer biology, to give a more complete picture of the role of these proteins in CSC and cancer biology.

### 3.3. Impact of GREM1 on Tumor Microenvironment

The tumor microenvironment niche plays a crucial role in regulating CSCs and their interactions with surrounding cells. An early publication on GREM1 and cancer described GREM1 mRNA expression in cancer-associated stromal cells, linked to promotion of tumor cell proliferation [38]. GREM1 is secreted by both cancer cells and stromal cells within the tumor microenvironment. GREM1 mRNA was detected in the cancer invasion front in a model of colorectal cancer, contributing to cancer cell motility [72]. The interaction between CSCs and stromal cells is vital for maintaining the CSC niche. Increased GREM1 levels can alter the composition of the tumor microenvironment, promoting the survival and proliferation of CSCs. Overexpression of GREM1 may facilitate CSC immune cell avoidance, as GREM1 has been shown to play a role in macrophage and dendritic cell function [73,74]. High levels of GREM1 were also detected in pancreatic stellate cells in the stroma of pancreatic tumors [58]. GREM1 expression in these cells was increased by Shh signaling, contributing to a fibrogenic stromal microenvironment and pancreatic cancer cell progression [54,58,75]. Expression of GREM1 from lung fibroblasts drove the proliferation of malignant lung adenocarcinoma cells [76]. Thus, GREM1 expression may play a role in both the “seed and soil” in a range of cancers, affecting the tumor stroma and the CSCs, creating a tumor-permissive environment for both primary tumors and metastases.

### 3.4. GREM1 as a Therapeutic Target in Cancer

Many reports have identified that high GREM1 levels correlate with poor patient prognosis in a range of human cancers [11,12,77,78]. Given its role in promoting CSC properties and promoting tumor growth and cancer progression, GREM1 represents an attractive therapeutic target for cancer treatment. Strategies aimed at inhibiting GREM1 expression or function could provide a means to not only target cancer cells but also disrupt the maintenance of CSCs and enhance chemotherapy effectiveness. Many companies have developed neutralizing monoclonal antibodies against GREM1. The first report detailing an anti-GREM1 neutralizing antibody was from Novartis, who showed that targeting GREM1 could reduce pulmonary artery hypertension in mice [79]. An in vivo study using a mouse model of multiple myeloma demonstrated that an anti-GREM1 neutralizing antibody from UCB reduced tumor burden by up to 81.2% [80]. In 2023, UCB launched a Phase 1/2 clinical trial to assess the safety, pharmacokinetics and anti-tumor activity of the GREM1 neutralizing antibody Ginisortamab (UCB6114) in patients with solid tumors in [81]. Preclinical data for Ginisortamab demonstrated that this antibody could reduce lung cancer cell proliferation in vitro [82]. Transcenta have developed TST003, a neutralizing GREM1 antibody that is currently in Phase 1 clinical trials, to test safety and pharmacodynamics in colorectal cancer patients with locally advanced or solid tumors [83]. Regeneron have also developed a GREM1 neutralizing antibody aimed at treating both fibrosis and cancer, and has been shown to attenuate osteoarthritis in a mouse model [84]. Given the discussions above, the potential of these neutralizing GREM1 antibodies to inhibit GREM1 signaling not only in cancer cells, but also in CSCs (thereby restoring BMP-induced CSC differentiation and other signaling), is a tantalizing prospect that could reduce the CSC population in tumors and improve patient outcomes.

In addition, small-molecule inhibitors that interfere with GREM1 interactions with BMPs and other signaling pathways are being developed by our group and others. For example, a compound called demethylenebernerine (DMB) has been shown to bind to deubiquitinase ubiquitin-specific peptidase 11 (USP11), which has been shown to regulate progression and chemoresistance of colorectal, breast, ovarian and other human cancers [85]. DMB treatment of a mouse model of pulmonary fibrosis led to disruption of GREM1 deubiquitination and GREM1 protein degradation [86]. Treatment with DMB compound was shown to alleviate pulmonary fibrosis in vivo, demonstrating that specific targeting of GREM1 in the lung may provide therapeutic benefit [86]. It is important to note that USP11 has a number of reported cellular substrates of USP11 apart from GREM1, including p21WAF1/CIP1 [87], cyclin D1 [88] and the Snail transcription factor [89]. Therefore, caution is needed when interpreting the effect of USP11 inhibitors and limiting the interpretation of in vivo data to a single USP11 substrate such as GREM1 should be avoided to ensure a full understanding of the likely effects of USP11 inhibition. Future development and optimization of cell-permeable GREM1 inhibitors is an exciting area of research that will help to inhibit both intra- and extracellular GREM1 signaling. These novel GREM1 inhibitors have the potential to contribute to chemotherapy strategies aiming to diminish the self-renewal and pluripotency of CSCs, enhancing chemotherapy effectiveness, and improving patient outcomes.

## 4. Conclusions

The role of GREM1 in human cancer is well detailed, with most of the research supporting a tumor promoter role for GREM1 overexpression in CRC and other human cancers. Regulation of CSC biology may also represent an important role for GREM1 in modulating human cancer. Again, most published research suggests that high levels of GREM1 represent a cellular mechanism by which CSCs can maintain their pluripotency, which presents a barrier to chemotherapy treatments aimed at the destruction of cancer cells. Future research will shed new light on the exact biology of GREM1 in CSC regulation, which will further accelerate pharmaceutical strategies aiming to disrupt GREM1 using neutralizing antibodies or small molecule therapeutics. The availability of novel anti-GREM1 pharmaceutics should expand and improve the treatment options for clinicians aiming not only to kill cancer cells and reduce tumor burden, but also to specifically target CSCs and lower the cancer recurrence risk. These exciting prospects have the potential to improve cancer patient outcomes over the next 10 years.

## Figures and Tables

**Figure 1 cells-14-00578-f001:**
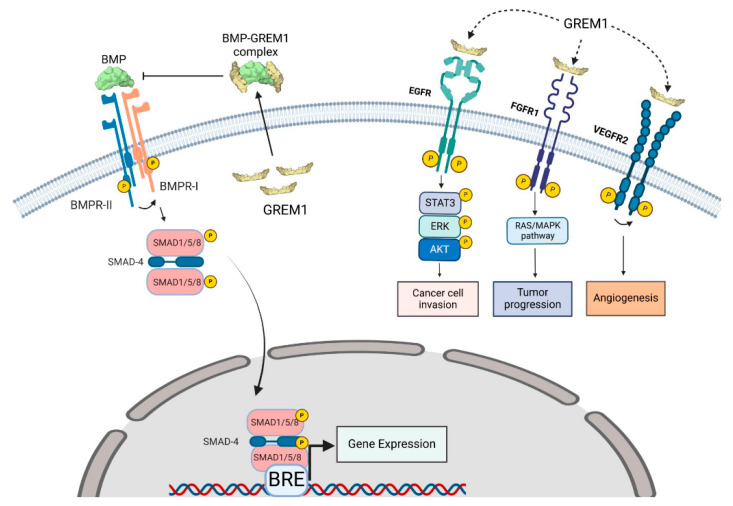
Schematic summarizing GREM1 signaling pathways. Binding of BMPs (green) to their cognate BMPRI/II complexes leads to receptor transphosphorylation, activation, and phosphorylation of R-Smad1/5/8 proteins. These proteins form complexes with Smad4 that then translocate to the nucleus and assemble with other elements of the transcription machinery at BMP-response elements (BRE) on BMP-responsive genes, leading to gene expression. Canonical GREM1 signaling involves secretion of covalently bound GREM1 dimers (gold) that bind and sequester BMPs in the extracellular matrix, preventing receptor activation and signaling. Other non-canonical signaling pathways that have been suggested for GREM1 dimers include EGFR activation leading to breast cancer cell invasion, FGFR1 activation leading to prostate cancer tumor progression, and VEGFR2 leading to angiogenesis. The dotted arrows indicate signaling pathways that require further validation. The complex interplay of all of these signaling pathways involving GREM1 requires further investigation. Image created in https://www.biorender.com/.

**Figure 2 cells-14-00578-f002:**
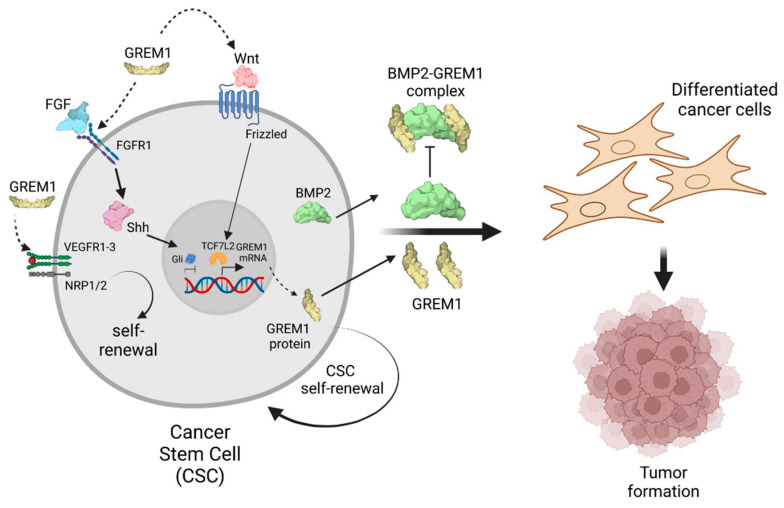
Regulation of CSCs by GREM1 signaling. Cancer stem cells (CSCs) express high levels of BMP2 (green) that drive CSC differentiation to cancer cells. However, CSCs also express high levels of GREM1 (gold) that bind to and inhibit BMP2, preventing differentiation and promoting CSC self-renewal and stemness. FGF signaling via FGFR1 → sonic hedgehog (Shh) activates the expression of Gli transcription factors (Gli1-3) that bind cis-regulatory elements at the *GREM1* promoter and inhibit GREM1 transcription. Wnt activation of Frizzled receptors leads to recruitment of TCF7L2 transcription factor to an enhancer element of the *GREM1* gene which increases GREM1 expression. The dotted arrow between GREM1 and Wnt represents data suggesting that GREM1 can modulate Wnt signaling in epithelial stem cells, enhancing self-renewal. Similarly, the dotted arrow between GREM1 and the FGFR1 reflects data suggesting that GREM1 can signal via FGFR1 to regulate prostate cancer cell lineage plasticity. VEGFR1-3 activation by VEGF (red ball) has been shown to regulate CSC self-renewal, involving the neuropilin 1/2 co-receptor (NRP1/2). A potential role for GREM1 in activating VEGFR2 and mediating CSC self-renewal is indicated by the dotted arrow. Image created in https://www.biorender.com/.

## Data Availability

Not applicable.

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
