# Peer review of "The Role of GREMLIN1, a Bone Morphogenetic Protein Antagonist, in Cancer Stem Cell Regulation"

_cells, 2025, doi:10.3390/cells14080578_

Round 1

Reviewer 1 Report (Previous Reviewer 3)

Comments and Suggestions for Authors

I am quite satisfied with the revisions of the manuscript, which was greatly improved compared to the previous version. (Some typos in the text should be corrected. For example, a period (.) is missing in the last sentence of the first paragraph on Page 8.)

Author Response

Thank you for the positive comments. We have corrected the typos and included the period in the last sentence of the 1st paragraph on page 8.

Reviewer 2 Report (Previous Reviewer 2)

Comments and Suggestions for Authors

It is appreciated that the authors have made some corrections in the revised version. However, the flow of the manuscript is still not satisfactory. For example, in Figure 2, it is unclear why the authors started discussing glioma tumors. If this graph is meant to elucidate the general role of GREM1 signaling in CSCs, shouldn't it apply to all types of cancer? Additionally, the two new figures lack detailed descriptions in the manuscript.

  1. Is there any specific reason the authors changed the title from “GREM1” to “GREMLIN1” but kept “GREM1” in the rest of the manuscript?
  2. The phrase “in health and disease” has not been corrected.
  3. "TGFβ" should be corrected to "TGF-β".
  4. There are some logical and grammatical mistakes throughout the manuscript; please read through carefully to correct them.
Comments on the Quality of English Language

The logic and grammar issues 

Author Response

It is appreciated that the authors have made some corrections in the revised version. However, the flow of the manuscript is still not satisfactory. For example, in Figure 2, it is unclear why the authors started discussing glioma tumors. If this graph is meant to elucidate the general role of GREM1 signaling in CSCs, shouldn't it apply to all types of cancer?

We have removed the reference to glioma tumors in the legend for Fig. 2 to comply with the reviewer's helpful suggestion.

Additionally, the two new figures lack detailed descriptions in the manuscript.

The detailed description for the figures is in the legends, and also in the text. We have referred the reader to the figures in the manuscript (Fig. 1, page 2 line 87-88; Fig. 2, page 3 line 161-162).

  1. Is there any specific reason the authors changed the title from “GREM1” to “GREMLIN1” but kept “GREM1” in the rest of the manuscript?

We used the full spelling of the gene in the title, and then abbreviated it upon first use in the abstract (page 1 line 17), and then used the abbreviated version (GREM1) in the rest of the manuscript, as per standard practice.

  1. The phrase “in health and disease” has not been corrected.

Apologies-we have now deleted this phrase in the abstract (page 1 line 29)

  1. "TGFβ" should be corrected to "TGF-β".

Corrected throughout

  1. There are some logical and grammatical mistakes throughout the manuscript; please read through carefully to correct them.

We have corrected these mistakes, and will do a final check of the accepted proofs-thank you.

Reviewer 3 Report (New Reviewer)

Comments and Suggestions for Authors

From my point of view, this review is innovative and well structured. In fact, all its sections are congruent with each other and with the title, i.e. the focus of the review, and each section is comprehensive and clear. The references are congruent. I believe that this review can make a good contribution to the scientific community.

Author Response

Thank you for the positive comments.

Reviewer 4 Report (New Reviewer)

Comments and Suggestions for Authors

The authors prepared detailed and well-written review manuscript. Role of GREM1 and BMP in cancer is thoroughly discussed.

Author Response

Thank you for the positive comments.

Round 2

Reviewer 2 Report (Previous Reviewer 2)

Comments and Suggestions for Authors

This second version is much improved, but some minor mistakes and logic issues remain. Please read carefully to correct them.

This manuscript is a resubmission of an earlier submission. The following is a list of the peer review reports and author responses from that submission.

Round 1

Reviewer 1 Report

Comments and Suggestions for Authors

The review comprehensively summarizes the roles of BMP and GREM1 in regulating the functions of cancer stem cells. It proposes inhibiting GREM1 may represent a novel therapeutic strategy for eliminating cancer stem cells and treating cancer. However, several areas within the manuscript require improvement or clarification.

major concerns

1. The sections "GREM1 and Cancer Stem Cells" and "Regulation of Stemness Maintenance by GREM1" are pivotal to discussing GREM1's relationship with cancer stem cells in this review. However, in the "GREM1 and Cancer Stem Cells" section, lines 153-186, the discussion pertinent to the high expression of GREM1 in cancer stem cells and its association with poor prognosis only extends to line 163. The subsequent content predominantly explores potential mechanisms of GREM1 overexpression in tumor cells, concluding with uncertainty about the applicability of these mechanisms to cancer stem cells. This appears to somewhat deviate from the main topic.

2. In the section "Regulation of Stemness Maintenance by GREM1," the author discusses how high GREM1 expression, resulting from chromosomal duplications, leads to HMPS. Clarifying the significant role of BMP inhibition in HMPS is essential here, as this segment of the discussion focuses on how GREM1 maintains stemness by inhibiting BMP.

3. The introduction specifically mentions colorectal cancer and dedicates a section to the role of BMP/GREM1 in the intestines, suggesting an intent to emphasize the function of GREM1 in the cancer stem cells of colorectal cancer. However, the rest of the document does not demonstrate that GREM1 has a more significant impact on colorectal cancer compared to other cancer types. This disconnect might lead to confusion about the specific focus of the review on colorectal cancer.

4. The section "GREM1 as a Therapeutic Target in Cancer" is overly simplistic. It would be beneficial to provide a more detailed discussion of drugs targeting GREM1 that are currently in preclinical studies or early clinical trials (Phase I/II).

5. The article does not utilize figures or tables, which hinders the reader's understanding of the content. Incorporating diagrams to illustrate the relationship between GREM1 and cancer stem cells and introduce relevant signaling pathways would significantly enhance the clarity of the discussion and improve the overall quality of the manuscript.

Minor concerns

1. Line 122 “that” repeat

2. Line 148 "EMT" and Line 165 "SNP" Define acronyms upon first use by providing the full term followed by the abbreviation in parentheses.

Author Response

Dear Editor,

Thank you for the correspondence regarding our commentary article entitled “The Role of GREMLIN1, a Bone Morphogenetic Protein Antagonist, in Cancer Stem Cell Regulation” by Gao et al. We value the comments from editors and reviewers alike, and have taken all advice on board to produce a revised, improved version of our article. We have detailed our changes below.

Editors Comments

The comments of the four reviewers show agreement on several unavoidable errors to be carried out in a revised version of the manuscript. The manuscript also shows problems of inconsistency in the objectives and purpose of the review. I believe that the manuscript cannot be reviewed and reedited but must be rejected and returned to the authors along with the reviewers' comments since these may be useful in the composition of future articles on the topic.

We have addressed most of these comments raised by the editor in the revised manuscript-specific improvements are listed below.

Reviewer 1

The review comprehensively summarizes the roles of BMP and GREM1 in regulating the functions of cancer stem cells. It proposes inhibiting GREM1 may represent a novel therapeutic strategy for eliminating cancer stem cells and treating cancer.

Thank you for the positive feedback.

However, several areas within the manuscript require improvement or clarification.

Major concerns

1.The sections "GREM1 and Cancer Stem Cells" and "Regulation of Stemness Maintenance by GREM1" are pivotal to discussing GREM1's relationship with cancer stem cells in this review. However, in the "GREM1 and Cancer Stem Cells" section, lines 153-186, the discussion pertinent to the high expression of GREM1 in cancer stem cells and its association with poor prognosis only extends to line 163. The subsequent content predominantly explores potential mechanisms of GREM1 overexpression in tumor cells, concluding with uncertainty about the applicability of these mechanisms to cancer stem cells. This appears to somewhat deviate from the main topic.

We have now edited this by expanding the Regulation of CSC Stemness Maintenance by GREM1 section (page 8) and moving some of the non-CSC text to the GREM1 Signaling in Cancer section as per the reviewer suggestion (page 6-7). We feel that it is important to present some of the data outlining the mechanisms by which GREM1 is overexpressed in cancer cells, which helps us introduce GREM1 overexpressing in CSCs.

  1. In the section "Regulation of Stemness Maintenance by GREM1," the author discusses how high GREM1 expression, resulting from chromosomal duplications, leads to HMPS. Clarifying the significant role of BMP inhibition in HMPS is essential here, as this segment of the discussion focuses on how GREM1 maintains stemness by inhibiting BMP.

We have added some text to clarify the role of BMP inhibition in HMPS as suggested (page 7).

  1. The introduction specifically mentions colorectal cancer and dedicates a section to the role of BMP/GREM1 in the intestines, suggesting an intent to emphasize the function of GREM1 in the cancer stem cells of colorectal cancer. However, the rest of the document does not demonstrate that GREM1 has a more significant impact on colorectal cancer compared to other cancer types. This disconnect might lead to confusion about the specific focus of the review on colorectal cancer.

We take this point on board and have made the introduction more general around the role of GREM1 in cancer. However, there is large volume of evidence that GREM1 may play a significant role in CRC, so we have alluded to this fact at the end of this section (page 3, 4).

  1. The section "GREM1 as a Therapeutic Target in Cancer" is overly simplistic. It would be beneficial to provide a more detailed discussion of drugs targeting GREM1 that are currently in preclinical studies or early clinical trials (Phase I/II).

We have expanded this section and added more detail about the preclinical studies and clinical trials (page 11, 12).

  1. The article does not utilize figures or tables, which hinders the reader's understanding of the content. Incorporating diagrams to illustrate the relationship between GREM1 and cancer stem cells and introduce relevant signaling pathways would significantly enhance the clarity of the discussion and improve the overall quality of the manuscript.

Due to time constraints for the original submission, we did not have time to generate figures. We now include 2 figures detailing GREM1/BMP signaling (Fig. 1) and a schematic summarising GREM1 signaling in CSCs (Fig. 2).

Minor concerns

1.Line 122 “that” repeat

Corrected.

  1. Line 148 "EMT" and Line 165 "SNP" Define acronyms upon first use by providing the full term followed by the abbreviation in parentheses.

Corrected.

Reviewer 2 Report

Comments and Suggestions for Authors

This manuscript attempts to review the role of GREM1, a Bone Morphogenetic Protein antagonist, in Cancer Stem Cells (CSCs). However, it lacks a clear focus and reads more like a collection of separate pieces without a coherent flow.

Major Issues:

  1. The authors specifically discuss Colorectal Cancer (CRC) in the introduction. Is GREM1 specifically related to this cancer type? If so, please elaborate on the relationship between GREM1 and CRC rather than just mentioning it generally. Also, emphasize this connection in the title. If not, reduce the emphasis on CRC mortality in the introduction and provide more information about the role of this molecule in other cancer types.
  2. In the introduction section, the first and second paragraphs can be combined as both discuss CRC mortality but from different scopes.
  3. The paragraph titled "BMP/GREM1 Signaling in the Intestine" does not fit well with the manuscript's context. It would be better to write a more general section, such as "GREM1 Signaling in Normal Tissues," which would be comparable to the following paragraph "GREM1 Signaling in Cancer."
  4. The sections "Characteristics of Cancer Stem Cells" and "Regulation of Stemness Maintenance by GREM1" might be better combined under one section.
  5. The sections "BMP Signaling and GREM1" and "BMP Inhibition" should be combined, as they cover similar content.
  6. It would be beneficial to elucidate the role of GREM1 in signaling pathways through a workflow, making the description more vivid.

Minor Issues:

  1. In Line 32, change "in health and disease" to "tumorigenesis" or another more appropriate term.
  2. In Line 225, correct "TGFβ" to "TGF-β" to maintain consistency with previous descriptions.
  3. Check citations throughout the manuscript to ensure each one is correctly cited.

Author Response

Dear Editor,

Thank you for the correspondence regarding our commentary article entitled “The Role of GREMLIN1, a Bone Morphogenetic Protein Antagonist, in Cancer Stem Cell Regulation” by Gao et al. We value the comments from editors and reviewers alike, and have taken all advice on board to produce a revised, improved version of our article. We have detailed our changes below.

Editors Comments

The comments of the four reviewers show agreement on several unavoidable errors to be carried out in a revised version of the manuscript. The manuscript also shows problems of inconsistency in the objectives and purpose of the review. I believe that the manuscript cannot be reviewed and reedited but must be rejected and returned to the authors along with the reviewers' comments since these may be useful in the composition of future articles on the topic.

We have addressed most of these comments raised by the editor in the revised manuscript-specific improvements are listed below.

Reviewer 2

This manuscript attempts to review the role of GREM1, a Bone Morphogenetic Protein antagonist, in Cancer Stem Cells (CSCs). However, it lacks a clear focus and reads more like a collection of separate pieces without a coherent flow.

Major Issues:

1.The authors specifically discuss Colorectal Cancer (CRC) in the introduction. Is GREM1 specifically related to this cancer type? If so, please elaborate on the relationship between GREM1 and CRC rather than just mentioning it generally. Also, emphasize this connection in the title. If not, reduce the emphasis on CRC mortality in the introduction and provide more information about the role of this molecule in other cancer types.

This is similar to Point 3 raised by Reviewer 1. We have broadened the text on the role of GREM1 in other forms of cancer and have added some specifics about GREM1 in CRC as per the reviewer’s suggestion (page 3, 4).

In the introduction section, the first and second paragraphs can be combined as both discuss CRC mortality but from different scopes.

Completed (page 3).

The paragraph titled "BMP/GREM1 Signaling in the Intestine" does not fit well with the manuscript's context. It would be better to write a more general section, such as "GREM1 Signaling in Normal Tissues," which would be comparable to the following paragraph "GREM1 Signaling in Cancer."

We have removed this section as suggested, and added a paragraph on GREM1 expression in healthy tissues at the start of the GREM1 Signaling in Cancer section (page 6).

The sections "Characteristics of Cancer Stem Cells" and "Regulation of Stemness Maintenance by GREM1" might be better combined under one section.

We have kept these sections separate as we have expanded the Characteristics of Cancer Stem Cells section, and believe that the article benefits from a dedicated Regulation of CSC Stemness Maintenance by GREM1 section, which we have also expanded to cover FGFàShh signaling, Wnt/Frizzled signaling and VEGFR signaling as mediators of GREM1 and CSC stemness (page 8-10).

The sections "BMP Signaling and GREM1" and "BMP Inhibition" should be combined, as they cover similar content.

The revised article now only has 1 section titled Regulation of BMP Signaling by GREM1. We have retained the paragraph on BMP inhibition by GREM1 in the Regulation of Stemness Maintenance by GREM1 as this is a key mechanism described in glioma.

It would be beneficial to elucidate the role of GREM1 in signaling pathways through a workflow, making the description more vivid.

We have included 2 new figures, and Fig. 1 highlights GREM1 signaling pathways to address the reviewer’s point.

Minor Issues:

  1. In Line 32, change "in health and disease" to "tumorigenesis" or another more appropriate term.

Done.

  1. In Line 225, correct "TGFβ" to "TGF-β" to maintain consistency with previous descriptions.

Done.

  1. Check citations throughout the manuscript to ensure each one is correctly cited.

Checked and completed.

Reviewer 3 Report

Comments and Suggestions for Authors

"The role of GREM1, a Bone Morphogenetic Protein antagonist, in Cancer Stem Cell regulation" summarized the recent progress on studying the mechanisms of GREM1 in CSC and its potential in translational medicine. It will be very interesting for the general audience of the journal.

The major problem is the inconsistency of the text. The authors spent many words describing colorectal cancer (CRC). However, CRC was mentioned very few times in the rest of the main text. So I recommend the authors to trim down CRC part.

In addition, the description of CSC (line 66 to 86) should be incorporated into the Introduction section instead of a separate paragraph.

The mechanisms of GREM1 in CSC (line 153 to 235) should be integrated into one section under one subtitle.

Author Response

Dear Editor,

Thank you for the correspondence regarding our commentary article entitled “The Role of GREMLIN1, a Bone Morphogenetic Protein Antagonist, in Cancer Stem Cell Regulation” by Gao et al. We value the comments from editors and reviewers alike, and have taken all advice on board to produce a revised, improved version of our article. We have detailed our changes below.

Editors Comments

The comments of the four reviewers show agreement on several unavoidable errors to be carried out in a revised version of the manuscript. The manuscript also shows problems of inconsistency in the objectives and purpose of the review. I believe that the manuscript cannot be reviewed and reedited but must be rejected and returned to the authors along with the reviewers' comments since these may be useful in the composition of future articles on the topic.

We have addressed most of these comments raised by the editor in the revised manuscript-specific improvements are listed below.

Reviewer 3

"The role of GREM1, a Bone Morphogenetic Protein antagonist, in Cancer Stem Cell regulation" summarized the recent progress on studying the mechanisms of GREM1 in CSC and its potential in translational medicine. It will be very interesting for the general audience of the journal.

Thank you for the positive feedback.

The major problem is the inconsistency of the text. The authors spent many words describing colorectal cancer (CRC). However, CRC was mentioned very few times in the rest of the main text. So I recommend the authors to trim down CRC part.

Similar to the feedback from other reviewers, we have taken the reviewer’s advice on board and reduced the focus on CRC throughout the article.

In addition, the description of CSC (line 66 to 86) should be incorporated into the Introduction section instead of a separate paragraph.

We have briefly introduced CSCs in the Introduction, and then expanded on this in the Characteristics of Cancer Stem Cells section (page 4).

The mechanisms of GREM1 in CSC (line 153 to 235) should be integrated into one section under one subtitle.

Completed-page 8-9.

Reviewer 4 Report

Comments and Suggestions for Authors

The manuscript does not contain Figures and/or Tables.

It is too premature whether regulation of GREM1 could be a promising therapeutic option for cancers.

Manuscript Review Comments

General Comments

  1. The reason why the authors selected GREM1 is not clear. There are many antagonists for BMPs, such as noggin, chordin, and follistatin.
  2. The proposed future implication of GREM1 for clinical application in cancer therapy lacks sufficient evidence. Therefore, it is premature to make such a statement.
    • What is the current status of GREM inhibitors as discussed in reference [60]?
    • What is the specificity of USP1 toward GREM1?
  3. The effects of GREM1 inhibitors on cancer stem cells remain unclear.

Specific Comments

  1. The manuscript lacks figures and tables, which diminishes its adequacy as a review article. Figures and /or Tables would enhance clarity and engagement.
  2. Line 12: What do the authors mean by "their"? Both BMP and GDF belong to the TGF-beta superfamily, so the pronoun usage should be clarified.
  3. Line 42: Does "the third" mean worldwide? Please specify.
  4. Line 70: Reference [29] is cited here, but the previous citation is [7]. Please ensure the references are numbered consistently.
  5. Line 92 The statement that GREM1 is an antagonist of the BMP subfamily. The statement that it is a member of the TGF-beta family is correct, and more precise phrasing is needed for clarity.

Author Response

Dear Editor,

Thank you for the correspondence regarding our commentary article entitled “The Role of GREMLIN1, a Bone Morphogenetic Protein Antagonist, in Cancer Stem Cell Regulation” by Gao et al. We value the comments from editors and reviewers alike, and have taken all advice on board to produce a revised, improved version of our article. We have detailed our changes below.

Editors Comments

The comments of the four reviewers show agreement on several unavoidable errors to be carried out in a revised version of the manuscript. The manuscript also shows problems of inconsistency in the objectives and purpose of the review. I believe that the manuscript cannot be reviewed and reedited but must be rejected and returned to the authors along with the reviewers' comments since these may be useful in the composition of future articles on the topic.

We have addressed most of these comments raised by the editor in the revised manuscript-specific improvements are listed below.

Reviewer 4

The manuscript does not contain Figures and/or Tables.

We have now added 2 Figures outlining the signaling mechanisms of GREM1, as well as the potential roles of GREM1 in CSC regulation (Fig. 1 and 2).

It is too premature whether regulation of GREM1 could be a promising therapeutic option for cancers.

Several pharma companies are developing neutralising antibodies targeting GREM1, and several of these are now in early-phase clinical trials. Therefore, we believe that it is reasonable to include a section on GREM1 targeting in this article. Similar to point 4 raised by Reviewer 1, we have expanded the section on GREM1 as a therapeutic target to provide more details and allow the reader to better understand the status of GREM1 targeting in cancer (page 11-13).

Manuscript Review Comments

General Comments

1. The reason why the authors selected GREM1 is not clear. There are many antagonists for BMPs, such as noggin, chordin, and follistatin.

There is a large and convincing body of literature implicated GREM1 as a bad actor in a wide range of human cancers. High levels of GREM1 are associated with poor patient prognosis for colorectal, breast, gastric and other cancers. Many pharma companies are developing anti-GREM1 therapeutics as potential anti-cancer drugs. While other antagonists such as follistatin and noggin also have important roles in cancer, there is little or no data on their role in regulating CSCs. The data on GREM1 in human cancer prompted a more focused review on GREM1 and the regulation of CSCs for this article. To address the reviewer’s point, we have added some text on the role of Noggin, Chordin, and Follistatin in CSC regulation and cancer (page 10).

 The proposed future implication of GREM1 for clinical application in cancer therapy lacks sufficient evidence. Therefore, it is premature to make such a statement.

  • What is the current status of GREM inhibitors as discussed in reference [60]?
  • What is the specificity of USP1 toward GREM1?

We have expanded this section to shed more light on the current status of GREM1 inhibitors (page 11-12) and addressing the specificity of USP1 toward GREM1 (page 12-13).

2. The effects of GREM1 inhibitors on cancer stem cells remain unclear.

We agree with this statement, and the effects of anti-GREM1 neutralizing antibodies and small molecule GREM1 inhibitors need to be fully characterized in both cancer cells and CSCs. We hope that the emerging availability of novel GREM1 inhibitors will allow us to test these compounds on a range of CSCs.

Specific Comments

1. The manuscript lacks figures and tables, which diminishes its adequacy as a review article. Figures and /or Tables would enhance clarity and engagement.

We have now added 2 figures to address this important comment.

2. Line 12: What do the authors mean by "their"? Both BMP and GDF belong to the TGF-beta superfamily, so the pronoun usage should be clarified.

“Their” in this context refers to cancer stem cells-apologies for confusion

3. Line 42: Does "the third" mean worldwide? Please specify.

We have removed this section as per advice from other reviewers.

4. Line 70: Reference [29] is cited here, but the previous citation is [7]. Please ensure the references are numbered consistently.

We have corrected this oversight.

5. Line 92 The statement that GREM1 is an antagonist of the BMP subfamily. The statement that it is a member of the TGF-beta family is correct, and more precise phrasing is needed for clarity.

We have changed this sentence to read “”.